# Cationic Pollutant Removal from Aqueous Solution Using Reduced Graphene Oxide

**DOI:** 10.3390/nano12030309

**Published:** 2022-01-18

**Authors:** Talia Tene, Stefano Bellucci, Marco Guevara, Edwin Viteri, Malvin Arias Polanco, Orlando Salguero, Eder Vera-Guzmán, Sebastián Valladares, Andrea Scarcello, Francesca Alessandro, Lorenzo S. Caputi, Cristian Vacacela Gomez

**Affiliations:** 1Grupo de Investigación Ciencia y Tecnología de Materiales, Universidad Técnica Particular de Loja, Loja 110160, Ecuador; tbtene@utpl.edu.ec; 2INFN-Laboratori Nazionali di Frascati, Via E. Fermi 54, I-00044 Frascati, Italy; Stefano.Bellucci@lnf.infn.it; 3School of Physical Sciences and Nanotechnology, Yachay Tech University, Urcuquí 100119, Ecuador; mvguevara@yachaytech.edu.ec; 4ITECA—Instituto de Tecnologías y Ciencias Avanzadas, Villarroel y Larrea, Riobamba 060104, Ecuador; 5Faculty of Mechanical Engineering, Escuela Superior Politécnica de Chimborazo, Riobamba 060155, Ecuador; eviteri@espoch.edu.ec; 6Instituto Tecnológico de Santo Domingo, Área de Ciencias Básicas y Ambientales, Av. Los Próceres, Santo Domingo 10602, Dominican Republic; melvin.arias@intec.edu.do; 7UNICARIBE Research Center, University of Calabria, I-87036 Rende (CS), Italy; orlando.salguero@yachaytech.edu.ec (O.S.); eder.vera@yachaytech.edu.ec (E.V.-G.); sebastian.valladares@yachaytech.edu.ec (S.V.); andrea.scarcello@unical.it (A.S.); francesca.alessandro@unical.it (F.A.); lorenzo.caputi@fis.unical.it (L.S.C.); 8Surface Nanoscience Group, Department of Physics, University of Calabria, Via P. Bucci, Cubo 33C, I-87036 Rende, Italy; 9INFN, Sezione LNF, Gruppo Collegato di Cosenza, Via P. Bucci, I-87036 Rende, Cosenza, Italy

**Keywords:** graphene oxide, reduce graphene oxide, dyes, heavy metals, pollutant removal

## Abstract

Reduced graphene oxide (rGO) is one of the most well-known graphene derivatives, which, due to its outstanding physical and chemical properties as well as its oxygen content, has been used for wastewater treatment technologies. Particularly, extra functionalized rGO is widely preferred for treating wastewater containing dyes or heavy metals. Nevertheless, the use of non-extra functionalized (pristine) rGO for the removal of cationic pollutants is not explored in detail or is ambiguous. Herein, pristine rGO—prepared by an eco-friendly protocol—is used for the removal of cationic pollutants from water, i.e., methylene blue (MB) and mercury-(II) (Hg-(II)). This work includes the eco-friendly synthesis process and related spectroscopical and morphological characterization. Most importantly, the investigated rGO shows an adsorption capacity of 121.95 mg g^−1^ for MB and 109.49 mg g^−1^ for Hg (II) at 298 K. A record adsorption time of 30 min was found for MB and 20 min for Hg (II) with an efficiency of about 89% and 73%, respectively. The capture of tested cationic pollutants on rGO exhibits a mixed physisorption–chemisorption process. The present work, therefore, presents new findings for cationic pollutant adsorbent materials based on oxidized graphenes, providing a new perspective for removing MB molecules and Hg(II) ions.

## 1. Introduction

Water pollution is one of the world’s most serious problems due to a large amount of wastewater being produced and poured into the water bodies every year [1]. Among different types of wastewaters, water contaminated with dyes and heavy metals deserves significant attention. With the continuous development of industrialization processes, large amounts of dyes or heavy metals are released into the environment, making water unsafe for human use and disrupting aquatic ecosystems [2,3,4]. A large amount of dye pollutants in the wastewater brings a huge risk to both aquatic organisms and humans because they can reduce sunlight transmission and normally contain toxic substances such as heavy metals [5], causing mutagenicity, carcinogenicity as well as the dysfunction of the kidney, liver, brain, reproductive system, and central nervous system [6]. On the other hand, agricultural processes or mining activities have increased the concentration of heavy metals in water around the world, which has led to immediate legislation by various governments [7]. When these toxic metal ions enter the food chain and then the human body, they accumulate in an organ above the allowed limits, originating serious health-related diseases, for instance: skin irritation, vomiting, stomach cramps, and cancer of the lungs and kidney [8]. Additionally, infants may have delays in physical or mental development, children may have deficits in attention span and learning activities, and adults may have high blood pressure [9].

To remove dyes or heavy metals, different physicochemical and electrochemical methods have been proposed. Physicochemical processes include membrane filtration [10], ion-exchange [11], and adsorption [12]. Electrochemical processes include electrocoagulation [13], electroflotation [14], and electrodeposition [15]. Among all these possible methods, including on-site sensing ones [16,17], those cost-effective, environmentally friendly and no further pollutant features are required. Therefore, adsorption is one of the most prominent approaches to water and wastewater decontamination. Due to this fact, many adsorbents with different structural conformations and compositions have been prepared or modified, for instance, clays/zeolites [18], biosorbents [19], agricultural solid wastes [20], and industrial by-products [21]. In practical operations, activated carbon is one of the most used adsorbents, thanks to its excellent adsorption performance [22]. However, activated carbon has been limited by the high cost and the complicated regeneration processes. To tackle the aforementioned limitations, carbon-based nanomaterials, such as carbon nanotubes [23], graphene oxide (GO) [24], and reduced graphene oxide (rGO) [25], have been proposed.

Recently, rGO has attracted increased interest as an effective adsorbent for dyes or heavy metal ions [25]. While the properties of rGO are quite different from those of pristine graphene or beyond-graphene materials (i.e., unique electronic, optical, mechanical, plasmonic, and thermal properties [26,27,28,29] as well as promising applications in hybrid capacitors [30]), rGO is characterized by interesting hydrophilic and semiconducting properties [31]. Nowadays, the synthesis of rGO follows an eco-friendly and cost-effective preparation method that can be used for large-scale water treatment technologies [32,33]. Moreover, the presence of oxygen functional groups (mainly hydroxyl and epoxide groups [34]) on the rGO surface allows covalent modifications with strong chelating groups, which, in turn, present a high affinity to metal ions or more complex organic/inorganic molecules.

There is extensive literature on the use of extra functionalized rGO or GO (e.g., GONR [35], S-GO [36], GO-TSC [37], S-doped g-C_3_N_4_/LGO [38], GSH-NiFe_2_O_4_/GO [39], HT-rGO-N [40]) for treating water and wastewater; however, the use of pristine rGO is scarce (and sometimes unclear) when removing cationic heavy metals and cationic dyes from aqueous media. In this work, such a comparative study is presented, considering methyl blue (MB) as a cationic dye and mercury (II) (Hg(II)) as cationic heavy metal. These cationic pollutants have been selected, in particular, because MB can cause various poisoning problems and methemoglobinemia [41] and Hg (II) can cause substantial neurodevelopmental risk in fetuses, newborns, and children [42]. Various kinetics, isotherms, and thermodynamic studies are carried out to demonstrate the adsorption of MB and Hg(II) on as-made rGO. Additionally, the present work includes the synthesis of adsorbent material and the corresponding morphological and spectroscopical characterizations of raw graphite, GO, and rGO.

## 2. Materials and Method

Although there are several methods for synthesizing graphene and its derivatives (e.g., liquid exfoliation [43,44], zeolite-assisted exfoliation [45], hydrothermal exfoliation [46], and microwave-assisted exfoliation [47,48,49,50,51,52,53]), the present study focuses on the eco-friendly oxidation-reduction protocol [33] to transform as-made GO into rGO.

### 2.1. Materials

All chemicals were used as received, without further purification:Graphite powder (<150 μm, 99.99%, Sigma-Aldrich, Burlington, MA, USA);Sulfuric acid (H_2_SO_4_, 95.0–98.0%, Sigma-Aldrich, Burlington, MA, USA);Potassium permanganate (KMnO_4_, ≥99.0%, Sigma-Aldrich, Burlington, MA, USA);Hydrochloric acid (HCl, 37%, Sigma-Aldrich, Burlington, MA, USA);Citric acid (C_6_H_8_O_7_, ≥99.5%, Sigma-Aldrich, Burlington, MA, USA);Methyl blue (MB, C_16_H_18_N_3_ClS, Sigma-Aldrich, Burlington, MA, USA);Hydrogen peroxide (H_2_O_2_, 30%, Merk, Darmstadt, Germany);Sodium hydroxide (NaOH, 1310-73-2, 40.00 g/mol, Merk, Darmstadt, Germany);Mercury (II) oxide (HgO, 21908-53-2, 219.59 g/mol, Merk, Darmstadt, Germany).

### 2.2. Preparation of Oxidized Graphenes

A round-bottom flask was charged with graphite (1.5 g), H_2_SO_4_ (35 mL), maintaining a uniform and moderate circular agitation. The mixture was located in an ice-water bath, and then KMnO_4_ (4.5 g) was slowly added. The resulting mixture was agitated on a stirring plate while adding 75 mL of distilled water, being careful not to exceed 363 K.

Additionally, 250 mL of distilled water was added, followed by 7.5 mL of H_2_O_2_. The resulting solution was distributed to be washed by centrifugation with HCl solution and distilled water several times to adjust the pH~6 [33] and then dried (drying stove, 60 Hz, 1600 W) at 353 K for 2 h to obtain graphite oxide flakes.

After the oxidation process, 50 mg of graphite oxide flakes were dispersed in 500 mL of distilled water by sonication for 0.5 h [25]. The resulting solution was centrifuged to separate GO from non-exfoliated graphite oxide particles [32]. Under agitation, 1.0 g citric acid (CA) was added to the centrifuged suspension, setting the reduction temperature at 368 K. The precipitated rGO was collected, washed with distilled water by centrifugation, and dried at 353 K for 2 h to obtain rGO powder [33]. The resulting rGO was used for the cationic pollutant removal.

### 2.3. Characterization of GO and rGO

The surface morphology of raw graphite, GO, and rGO were carried out on a transmission electron microscope (TEM, JEM 1400 Plus, Akishima, Tokyo, Japan) operating at 80 kV, and a scanning electron microscope (SEM, JSM-IT100 InTouchScope, Akishima, Tokyo, Japan) equipped with a JEOL dispersive X-ray spectrometer (EDS) (Billerica, MA, USA), with the accelerating voltage of 15 kV. Raman spectra of graphite and oxidized graphene were obtained using a Jasco NRS-500 spectrometer (Oklahoma City, OK, USA), with a 532 nm laser wavelength (0.3 mW, 100X objective). Infrared spectra were collected using a Fourier transform infrared spectrometer (Jasco FT/IR 4000, Oklahoma City, OK, USA). UV-visible measurements were recorded using a UV–vis spectroscopy (Thermo Scientific, Evolution 220, Waltham, MA, USA). X-ray diffraction measurements were performed using an X-ray diffractometer (PANalytical Pro X-ray, Malvern, UK) in the diffraction angle (2θ) window of 5–90° using Cu Kα irradiation under the acceleration voltage of 60 kV and a current of 55 mA. The thermal stability of GO and rGO was examined using thermogravimetric analysis (TGA, PerkinElmer simultaneous thermal analyzer, STA 6000, Waltham, MA, USA).

SEM samples were mounted on aluminum substrates with adhesive, coated with 40–60 nm of metal such as Gold/Palladium and then observed in the microscope. TEM samples were arranged by drop-casting onto formvar-coated copper grids once the samples were cut into very thin cross-sections, allowing electrons to pass directly through the sample. Similarly, Raman samples were deposited directly by drop-casting on glass substrates and dried for a few seconds with the incident beam. To record the UV-visible spectra in the window range 190–1000 nm, the samples were redispersed in distilled water by mid-sonication for 5 min.

### 2.4. Preparation of MB Solutions and Experimental Set-Up of MB Adsorption on rGO

MB was dissolved in ultra-pure water to obtain a stock solution of 1000 mg L^−1^, and the working solutions were used in the test through serial dilutions. The pH of the solutions was adjusted using HCL and NaOH and controlled by a pH meter (HI221 Hanna Instruments).

The adsorption experiments were carried out in triplicate. In total, 500 mg of rGO was added into 250 mL of the MB solution with a concentration of 100 mg L^−1^ to evaluate the adsorption kinetics and contact time effect (batch test) [25]. The resulting mixture was agitated up to 60 min at 298 K. Adsorption isotherms were obtained from batch experiments by adding 200 mg rGO in 50 mL of MB solutions, considering different concentrations in the range of 10–100 mg L^−1^ and three different temperatures (298, 313, and 333 K). pH studies were carried out by adding 200 mg rGO in 100 mL of MB solution with a concentration of 100 mg L^−1^. Various aliquots were extracted from the solution to be evaluated by UV–vis spectroscopy. In all adsorption experiments (except in the study of the effect of pH), the pH was fixed to 6.02±0.07. To investigate the pH effect, the pH of MB solutions was adjusted by HCl (0.1 M) and NaOH (0.1 M), and immediately, rGO was added.

### 2.5. Preparation of Hg(II) Solutions and Experimental Set-Up of Hg(II) Adsorption on rGO

rGO was placed in a dilute aqueous solution of HgO. The adsorption kinetics studies were performed by adding 300 mL aqueous HgO (pH = 6.41±0.05) to a falcon tube. Then, 200 mg rGO was added to form a slurry. The mixture was stirred at 298 K for 60 min. After that, the mixture was filtered at intervals through a 0.45 mm membrane, and then the filtrated samples were analyzed by using an AAS-cold vapor to determinate the remaining Hg(II) content (standard methods 3112-B; 3111-B.4b) [13]. Adsorption isotherms were obtained by adding 2.5 mg rGO to each falcon tube containing 50 mL of HgO solution with different concentrations from 10 to 100 mg L^−1^ and considering three different temperatures (298, 313, and 333 K). The resulting mixtures were stirred at room temperature for 30 min and then filtered separately through a 0.45-mm membrane filter. The filtrates were analyzed by using AAS-cold vapor to determine the remaining Hg(II) content (standard methods 3112-B; 3111-B.4b) [13]. The pH effect was investigated by adjusting the pH of HgO solutions with HCl (0.1 M) and NaOH (0.1 M) at room temperature, and immediately, rGO was added.

## 3. Results and Discussion

### 3.1. Characterization of Graphite

Graphite has a crystal structure made up of stacked graphene layers in which the separation distance of the layers is 3.35 Å, whereas the separation of atoms within a layer is 1.42 Å. At the microscale, the starting (powder) graphite shows an irregular bulk structure with a lateral size ranging from 2 μm to 50 μm (Figure 1a). The XRD pattern of raw graphite is shown in Figure 1b. The most intense peak at 2θ=26.73° corresponds to the graphite stacking crystallinity (002) [43]. The less intense peak at 2θ=55.82° displays the long-range order of stacked graphene layers (004) [43].

Figure 1c,d show the Raman spectrum of raw graphite. The main features observed are: (i) the absence of the D peak demonstrating a defect-free starting graphite, (ii) the G peak at 1577 ${\mathrm{cm}}^{-1}$ is ascribed to the C-C strC stretching mode in sp^2^ carbon bonds, and (iii) the 2D peak at 2720 ${\mathrm{cm}}^{-1}$ is characterized by two bands, the intense 2D_2A_ band at 2720 ${\mathrm{cm}}^{-1}$ and a shoulder 2D_1A_ band at 2677 ${\mathrm{cm}}^{-1}$. In particular, these bands originate as the effect of the splitting of π electron bands due to the interaction between stacked graphene layers. The G* peak found at 2447 ${\mathrm{cm}}^{-1}$ is characteristic of carbon-based materials with a graphitic-like structure.

Compared to graphite, the Raman spectrum of oxidized graphenes shows a highly broadened and very-low intense 2D peak. With this in mind, the 2D band region of GO and rGO is not analyzed here, and instead, we focus on the region from 1000 to 2000 cm^−1^ to scrutinize the crystallinity and, most importantly, the basal/edge defects of the obtained materials after the oxidation-reduction process (discussed below).

### 3.2. Characterization of GO and rGO

While, in the present work, rGO is used for the adsorption of cationic pollutants, it is extremely important to discuss its transformation from GO. SEM micrographs of GO and rGO are shown in Figure 2a,c, respectively. The surface morphology of GO indicates a face-to-face stacking of flakes as well as randomly aggregated flakes with wrinkles and folds on the surface (Figure 2a). Instead, rGO shows a surface morphology with mesopores and micropores and randomly organized flakes (Figure 2c). The highly distorted porous surface of rGO can avoid the face-to-face stacking of flakes, as observed in GO.

EDS measurements were carried out to determine the elemental composition of GO and rGO, considering a bombarded region large enough. Then, the carbon and oxygen content were C: 49.7% and O: 50.3% for GO (after oxidation process) and C: 62.9% and O: 37.1% for rGO (after de reduction process). The oxygen content decreased by 26.2% using CA as an alternative green-reducing agent, confirming the (partial) removal of oxygen functional groups.

Representative TEM graphs of GO and rGO are shown in Figure 2e,f, respectively. GO looks like a semi-transparent thin nanosheet with various wrinkles and folds on the surface and edges (Figure 2e). The observed wrinkled/folded structure is attributed to surface defects because of the deviation from sp^2^ to sp^3^ hybridization as the effect of a high density of oxygen-containing functional groups [54]. After the reduction process with CA, well-defined and impurity-free nanosheets with slightly wrinkled regions are observed in rGO, suggesting the recovery of sp^2^ hybridization by the removal of functional groups. The observed regular surface allows concluding that rGO did not undergo severe in-plane disruption compared to GO.

Raman analyses were performed to further corroborate the transformation of GO into rGO (Figure 3a,b, respectively). As is typical for oxidized graphenes, two characteristic peaks are observed in GO and rGO: (i) the D peak at ~1349 cm^−1^ is attributed to the breathing mode of aromatic carbon rings, which is Raman active by structural defects [32], and (ii) the G peak at ~1588 cm^−1^ is due to the C-C stretching mode in the sp^2^ hybridized carbon structure [46]. A detailed analysis using Lorentz functions shows the existence of three prominent bands: the D band (yellow line), the G band (green line), and the D’ band (blue line). In particular, the D’ band confirms the presence of basal/edge defects, and a decrease in the D’ band intensity is a direct indication of GO reduction, which is observed in the Raman spectrum of rGO (Figure 3b). On the other hand, the I_D_/I_G_ intensity ratio can be used as an indicator of the density of structural defects in the obtained oxidized graphenes [43]. It was found that the intensity ratio of GO (2.2) is larger than that of rGO (1.65), indicating that the size of the graphene-like domains increases after the reduction process.

The absorbance spectra of GO and rGO are shown in Figure 3c,d, respectively. Using the Lorentzian function, GO has the main absorbance band at 230 nm (darker green line) and a shoulder band at 329 nm (yellow line), which are related to the $\pi -\pi^{*}$ transitions of C-C bonds and $n-\pi^{*}$ transitions of C=O bonds, respectively. To confirm the transformation of GO into rGO, two characteristics are needed: (i) a redshift of the main absorbance band and (ii) the loss of the shoulder band. After the reduction process, rGO only meets the first point when the main absorbance band shifts to 261 nm, but the second one is observed at 324 nm, suggesting a close content of oxygen-containing functional groups, particularly hydroxyl and epoxide groups.

The presence and type of oxygen functional groups are confirmed by the FTIR analysis (Figure 4a). It is widely accepted that the hydroxyl and epoxide groups are attached to the basal in-plane of the graphene, whereas the carboxyl and carbonyl groups are located at the edges. The FTIR spectrum of GO shows the following bands: C-O-C at 1044 ${\mathrm{cm}}^{-1}$, C-O at 1222 ${\mathrm{cm}}^{-1}$, C=C at 1644 cm^−1^, and C=O at 1729 ${\mathrm{cm}}^{-1}$. The broadband observed at ~3426 ${\mathrm{cm}}^{-1}$ is due to the presence of the hydroxyl groups (C-H) as well as adsorbed water molecules between GO flakes. The latter provides a hydrophilic characteristic in GO to be highly dispersible in water. It is worth noting that a higher hydrophilic property could interfere with the removal of pollutants from aqueous media, giving a poor adsorption process. After the reduction, these bands are significantly attenuated and weakened in the rGO spectrum, evidencing the removal of oxygen-containing functional groups [33].

To determine the thermal stability of as-made oxidized graphenes and the effect on the oxygen-containing functional groups, we carried out TGA analyses on GO and rGO (Figure 4b). In GO, the weight loss below 100 °C is ascribed to the loss of water molecules [33]. The significant weight loss in the region of 200–300 °C is attributed to the pyrolysis of unstable molecules (such as CO, CO_2_, and H_2_O) [33]. In the region of 300–600 °C, the weight loss is due to the removal of stable oxygen functional groups [33]. Instead, rGO shows relative thermal stability, but the observed TGA curve follows a similar trend as GO, confirming a reduced density of oxygen functional groups.

Finally, the crystallinity changes from GO to rGO were revealed by XRD analysis (Figure 4c). As mentioned, graphite is characterized by an intense crystalline peak at 2θ = 26.73° related to a lattice spacing of 0.334 nm, which corresponds to the (002) interplane distance [43] (Figure 1b). In GO, this peak is found at 2θ = 10.93° with a lattice spacing of 0.81 nm, indicating the oxidation of graphite. The increased interlayer spacing appears as an effect of the intercalation of water molecules and oxygen functional groups. Additionally, the very low width of this peak demonstrates an ordered stacking along the out-of-plane axis. After the reduction process, the peak becomes broader due to the partial breakdown of the long-range order, and it shifts towards higher angles, 2θ = 22°, showing a decrease in the lattice spacing (~0.39 nm) [43].

All these facts and pieces of evidence demonstrate the transformation of GO into rGO, which will be used for the removal of cationic pollutants from aqueous media, i.e., MB and Hg(II).

### 3.3. Adsorption Kinetics

We begin analyzing the effectiveness of rGO for removing MB and Hg(II) from water by using the following expression:(1)$$q_{t}=\frac{(C_{0}-C_{t})V}{W}$$
where $C_{0}$ and $C_{t}$ are the initial pollutant concentration (mg L^−1^) and the pollutant concentration at time t, respectively. W is the adsorbent mass (g), and V represents the volume of the aqueous solution (L). At the equilibrium, the equilibrium concentration is $C_{e}=C_{t}$, and the equilibrium adsorption capacity is $q_{e}=q_{t}$.

The removal efficiency (RE%) of the as-made rGO material can be defined by the following simple equation:(2)$$RE\%=\frac{\left(C_{0}-C_{e}\right)}{C_{0}}\times 100$$

Figure 5 shows the adsorption kinetics of MB and Hg (II) onto rGO at 298 K considering a contact time of up to 60 min. It can be seen that rGO rapidly captures MB molecules after 30 min (Figure 5a), while for Hg (II), the equilibrium time of adsorption is 20 min (Figure 5b). These results highlight the effectiveness of rGO for removing cationic pollutants from aqueous solutions compared with conventional benchmark sorbents [35,36,37,38,39,40]. In particular, the effectiveness of rGO can be attributed to the recovered surface area after the reduction process as well as the presence of oxygen functional groups.

The parameters of the adsorption kinetic process were determined by the pseudo-first-order model and pseudo-second-order model. Specifically, Tene et al. stated that the first model assumes that the rate of change of the adsorption capacity is proportional to the concentration of available active sites per unit mass of adsorbent material [25], whereas Arias et al. stated that the second model assumes that the rate of change of the concentration of occupied active sites per unit mass of the adsorbent material is proportional to the square of the concentration of free active sites per unit mass of sorbent [13].

The pseudo-first-order model (red line) and pseudo-second-order model (blue line) can be described as follows:(3)$$\log\;\left(q_{e}-q_{t}\right)=\log q_{t}-\frac{k_{1}}{2.303}t$$
and
(4)$$\frac{t}{q_{t}}=\frac{1}{k_{2}q_{e}^{2}}+\frac{1}{q_{e}}t$$
where $k_{1}$ and $k_{2}$ are the pseudo-first-order and pseudo-second-order rate constants, respectively. The estimated parameters of the adsorption kinetics are summarized in Table 1.

A close picture of the pseudo-first-order and pseudo-second-order parameters shows that, in the case of MB, the calculated values of the equilibrium adsorption capacity ($q_{e\left(cal\right)}=69.82$ mg g^−1^ and $q_{e\left(cal\right)}=70.72$ mg g^−1^, respectively) are very close to the experimental value ($q_{e\left(exp\right)}=68.21$ mg g^−1^). In the case of Hg(II), the calculated adsorption capacity ($q_{e\left(cal\right)}=143.71$ mg g^−1^) from the pseudo-first-model is close enough to the experimental value ($q_{e\left(exp\right)}=142.26$ mg g^−1^). However, the pseudo-second-order model overestimates the equilibrium adsorption capacity ($q_{e\left(cal\right)}=151.32$ mg g^−1^). By the comparison of SSE and R^2^ metrics, the adsorption kinetics of MB onto rGO are more in line with the pseudo-second-order model (SSE = 5.24, R^2^ = 0.999), whereas the adsorption kinetics of Hg(II) onto rGO are more in line with the pseudo-first-order model (SSE = 1826, R^2^ = 0.949).

### 3.4. Intraparticle Diffusion Study

The diffusion process of any pollutant into porous solid materials, such as our rGO (Figure 2c), mostly involves several steps characterized by different rates. This fact can be calculated by the intraparticle diffusion (IPD) model [13,25], which is given by the following expression:(5)$$q_{t}=k_{p}t^{0.5}+C$$
where $k_{p}$ is the intraparticle diffusion rate constant (g mg^−1^ min^−1^), and the intercept C reflects the boundary layer or surface adsorption. The respective plot and estimated parameters of the IPD model are shown in Figure 6 and Table 2.

As Ofomaja et al. [55] stated, the larger the intercept value, the greater the contribution of the surface in the adsorption process. Indeed, the values observed in MB (C=59.17) and Hg(II) (C=44.28) indicate that a greater amount of surface adsorption occurred, leading to a decrease in the rate of diffusion of MB molecules and Hg(II) ions from the adsorbent external surface to the adsorbent internal structure. From the linearized plot of the IPD model, different regions are observed: (i) the initial region (faster stage) is related to the movement of the pollutant from the solution to the rGO surface, (ii) the second region (intermediate stage) is related to the gradual diffusion of the pollutant into the large pores of the rGO structure, and (iii) the final region (lower stage) involves a very slow diffusion of the pollutant from larger pores to smaller ones.

Interestingly, the adsorption mechanism of MB on rGO is characterized by only two regions, regions I and II (Figure 6a), while all three regions are observed when Hg(II) becomes adsorbed onto rGO. In light of understanding this fact, we hypothesize that the size of pollutants plays a significant role in the diffusion procedure, i.e., as the MB molecules exhibit larger sizes compared to Hg(II) ions, MB cannot reach region III, particularly from larger to smaller pores.

The initial adsorption factor ($R_{i}$) can be estimated to further understand the above-mentioned regions (Table 2) as follows:(6)$$R_{i}=\frac{q_{ref}-C}{q_{ref}}$$
where $q_{ref}$ is the final adsorption amount at the longest time. In the MB-rGO system, the estimated $R_{i}$ value is much less than 0.5, which confirms that most of the adsorption of MB occurs on the surface of rGO. In contrast, for the Hg(II)-rGO system, the value of $R_{i}\sim$0.49 indicates a limit between the strong initial adsorption (related to region I) and intermediate initial adsorption (related to region II), which means that the adsorption process of Hg(II) ions could occur at almost the same time in both regions.

### 3.5. Adsorption Isotherms

Adsorption isotherms were carried out to analyze the interaction between MB molecules or Hg(II) ions and rGO considering a contact time of 30 min for MB and 20 min for Hg(II). The experimental data can be fitted using the Langmuir model and Freundlich model using the following equations, respectively:(7)$$q_{e}=\frac{q_{m}\;K_{L}\;C_{e}}{1+K_{L}\;C_{e}}$$
and
(8)$$q_{e}=K_{F}C_{e}^{1/n}$$
where $q_{m}$ represents the maximum adsorption capacity (mg g^−1^), $K_{L}$ is the Langmuir constant (L g^−1^), $K_{F}$ is Freundlich constant (mg L^−1^), and n is the surface heterogeneity of adsorbent material. The corresponding results and estimated parameters at different temperatures (298, 313, 333 K) are shown in Figure 7 and Figure 8 and Table 3 and Table 4.

Taking the high correlation R^2^ values (Table 3 and Table 4), it can be seen that the measured points are more in line with the Langmuir model. Although the temperature does not dramatically modify the chemical composition of rGO at temperatures below 100 °C (Figure 4b), it seems to be an important parameter in the adsorption process because when the temperature increases, in the case of MB on rGO, a slight decrease in the maximum adsorption capacity is observed from 121.95 to 107.53 mg g^−1^. In contrast, in the case of Hg (II) on rGO, a significant increase in the maximum adsorption capacity is detected from 109.49 to 255.04 mg g^−1^. The temperature is a key point to be considered if rGO is used to treat water or wastewater at an industrial scale.

From the Freundlich model, the estimated values of n for MB-rGO (Table 3) or Hg(II)-rGO (Table 4) systems indicate that the adsorbent heterogeneity tends to be homogeneous as the temperature rises. Indeed, values of n close to zero (<0.1) indicate strong surface heterogeneity. The affinity of the tested cationic pollutants for rGO can also be determined by the $K_{L}$ parameter, where the estimated values were found to be much less than 0.1, suggesting a good affinity of rGO to capture cationic pollutants, i.e., MB molecules and Hg(II) ions. However, this statement motivates more extended work for testing more cationic and non-cationic pollutants.

### 3.6. Effect of pH and Initial Concentration

To scrutinize the effect of the pH on the process of cationic pollutant removal, the experiments were carried out at different pH values ranging from 2 to 12 and setting the temperature at 298 K.

For MB on rGO (Figure 9a), the adsorption increases, starting from a removal percentage of about 76% at pH = 3 up to 92% at pH = 6. The removal percentage remains relatively constant from pH = 6 to pH = 8. The removal percentage decreases down to 83% for pH ≥ 10. To understand this fact, the effect of pH can be divided into three different regions: (i) from pH = 2 to pH = 4, the acid region is rich in cations which are captured together with the cationic MB molecules; (ii) the (relatively) neutral region from pH = 6 to pH = 8 is free from cations in the medium, and therefore, only the cationic dye molecules are captured by rGO; and (iii) the basis region (pH ≥ 10) is characterized by an excess of OH^−^ ions that interact with the cationic dye molecules, remaining suspended in the aqueous media.

For Hg(II) on rGO (Figure 9b), a removal percentage of about 39% at pH = 2 is observed, and the maximum removal percentage is found at pH = 10 (~82%). The removal percentage remains relatively constant for pH ≥ 6, with an average value of 76.58%. To understand these results, a similar description can be given: (i) for pH ≤ 4, the cations in the acidic medium fight with the mercury cations for the active sites of rGO; (ii) in the neutral region, mercury cations easily reach the active sites of rGO; and, interestingly, (iii) for pH ≥ 8, mercury cations sometimes prefer to interact with the active sites of rGO rather than OH^−^ ions due to the variation of the removal percentage when the pH increases.

The adsorption capacity ($q_{e}$) of rGO increases quite linearly with the initial concentration of MB in solution ($C_{0}$), almost in the range from 10 to 80 mg L^−1^; however, at high concentrations (≥90 mg L^−1^), a deviation from linearity does occur (Figure 10a). Similarly, the $q_{e}$ values of rGO increase linearly with the initial concentration of Hg(II) in the solution, from 10 to 50 mg L^−1^, and at concentrations ≥ 50 mg L^−1^, a deviation from linearity is also observed (Figure 10b). These results suggest that rGO has a finite amount of active adsorbent sites, which is fixed by its quality and the experimental conditions, such as temperature, pH, and solution volume/adsorbent mass ratio. To further emphasize, at the beginning of the adsorption process, rGO has a vast number of active sites, increasing the $q_{e}$ value as long as free active sites are available on rGO. Then, if all the active sites are involved, the saturation, and therefore the maximum adsorbent capacity ($q_{m}$), is attained [13,25].

The adsorption effectiveness of rGO—defined as the percentage of cationic pollutant removal from water—is almost independent of $C_{0}$ in the adsorption of MB onto rGO, assuming an average value of 89.21%. Interestingly, a clear dependence on $C_{0}$ is observed for the adsorption of Hg(II) onto rGO, i.e., an abrupt drop from 92.89% ($C_{0}=30$ mg L^−1^) to 48.85% ($C_{0}=100$ mg L^−1^), giving an average value of mercury removal of 72.93%.

### 3.7. Adsorption Thermodynamics

To acquire information about the energy changes due to the involved adsorption process [13], the Gibbs free energy ($∆G^{0}$), enthalpy change ($∆H^{0}$), and entropy change ($∆S^{0}$) were calculated by the following expressions:(9)$$K_{d}=\frac{q_{e}}{C_{0}}$$
(10)$$\ln K_{d}=\frac{∆S^{0}}{R}-\frac{∆H^{0}}{R\;T}$$
(11)$$∆G^{0}=-R\;T\ln K_{d}$$
where $K_{d}$ represent the distribution coefficient [13]. $∆H^{0}$ and $∆S^{0}$ were calculated from the slope and intercept of Van’t Hoff plot of $\ln K_{d}$ as a function of $T^{-1}$ [25]. The Van’t Hoff plot and estimated parameters are shown in Figure 11 and Table 5, respectively.

The negative $∆G^{0}$ values observed at different temperatures indicate spontaneous adsorption of MB molecules and Hg(II) ions onto the rGO surface. It is worth noting that, for $∆G^{0}$ values in the range from 0 to −20 kJ mol^−1^, the adsorption process is assigned to physisorption or multilayer adsorption [25], while in the range from −80 to −400 kJ mol^−1^, the adsorption is assigned to chemisorption or monolayer adsorption [25]. The region from −20 to −80 kJ mol^−1^ remains unclear, and a combined adsorption process can be assumed. With this in mind, the estimated $∆G^{0}$ values for MB onto rGO (−22.75, −23.81, and −25.16 kJ mol^−1^) and Hg(II) onto rGO (−39.84, 31.55, 32.97 kJ mol^−1^) suggest that the adsorption process of tested cationic pollutants on rGO is governed by a mixed physisorption–chemisorption process. Interestingly, for MB on rGO, the $∆G^{0}$ value increases by 5% at 313 K and by 10% at 333 K. In contrast, an inversely proportional relationship is observed for Hg(II) on rGO; say, the $∆G^{0}$ value decreases by 21% at 313 k and by 17% at 333 K.

The negative $∆H^{0}$ values indicate the exothermic nature of the adsorption process, i.e., a negative enthalpy implies that the temperature increase had a negative impact, particularly on the adsorption of MB ($∆H^{0}=-2.20$ kJ mol^−1^). However, in the adsorption of Hg(II) on rGO, the value observed ($∆H^{0}=-0.14$ kJ mol^−1^) is very small and could be considered negligible since increasing the temperature significantly increases the maximum adsorption capacity of rGO, as evidenced by the Langmuir model (Figure 8a and Table 4). The positive values of $∆S^{0}=0.069$ kJ mol^−1^ · K^−1^ and $∆S^{0}=0.079$ kJ mol^−1^ · K^−1^ corroborate the affinity of MB molecules and Hg(II) ions toward the rGO surface.

### 3.8. Final Remarks

Table 6 shows the estimated $q_{m}$ values for MB ($q_{m}=121.95$ mg g^−1^) and Hg(II) ($q_{m}=109.49$ mg g^−1^) at 298 K, which are compared to those of recent studies.

The estimated $q_{m}$ value of the dye adsorption is higher than those previously reported and only surpassed by GO reduced by Citrus hystrix ($q_{m}=276.06$ mg g^−1^), suggesting that as-made (non-extra functionalized) rGO are excellent platforms to replace conventional adsorbent materials. In the case of heavy metal adsorption, the estimated $q_{m}$ value is higher than some functionalized/decorated GO and rGO. However, S-GO seems to be more profitable to be used for the removal of mercury ($q_{m}=3490$ mg g^−1^), but this is due to the fact that, obviously, the presence of sulfur improves the affinity and specificity for Hg (II) ions in any adsorbent material.

Although, in the present work, the regeneration of rGO was not studied, which motivates more extended work, we propose the following well-known techniques or processes: (i) the adsorbed rGO-pollutant system can be separated from aqueous media by filtration using filters with a pore size less than 1 µm since rGO is within the order of few micrometers, (ii) pollutant can be released from rGO by applying the concept of ionic force, i.e., by applying buffer solutions, and (iii) the isolated MB molecules or Hg(II) ions can be extracted by sulfide precipitation.

## 4. Conclusions

In summary, we have demonstrated the effective and efficient removal of cationic pollutants (i.e., MB molecules and Hg(II) ions) from aqueous solutions using an eco-friendly and as-made rGO. The adsorbent material shows fast adsorption with a saturation capacity of 121.95 mg g^−1^ and 109.49 mg g^−1^ at 298 K, suggesting a good affinity for MB molecules and Hg(II) ions. These results are superior to those recently reported for other graphene-based benchmark materials [35,36,37,38,39,40]. By means of several chemical physics analyses, we have also shown that rGO keeps a good efficiency over a wide range of initial cationic pollutant concentrations and a broad range of pH values. Specifically, the maximum removal percentage as a function of pH was found in the range of 6 to 8 for MB and 6 to 10 for Hg(II). Our results allowed us to conclude that the MB-rGO and Hg(II)-rGO adsorption interaction follows a combined physisorption–chemisorption process due to the fact that the Gibbs free energy was found from −22.75 to −25.16 kJ mol^−1^ for MB and from −39.84 to −32.97 kJ mol^−1^ for Hg(II). The present study proposes non-extra functionalized rGO as a potential green adsorbent for wastewater decontamination.

## Figures and Tables

**Figure 1 nanomaterials-12-00309-f001:**
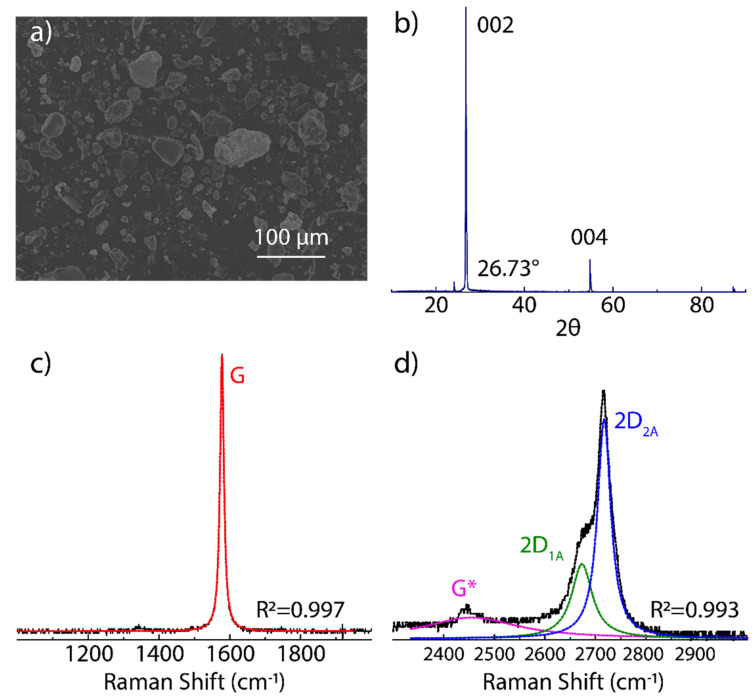
Characterization of starting graphite source: (**a**) SEM morphology, (**b**) XRD measurement, and (**c**,**d**) Raman spectrum from 1000 to 3000 ${\mathrm{cm}}^{-1}$ recorded using 532 excitation laser. The intensity was normalized by the most intense peak. The Raman spectrum was fitted using Lorentzian functions.

**Figure 2 nanomaterials-12-00309-f002:**
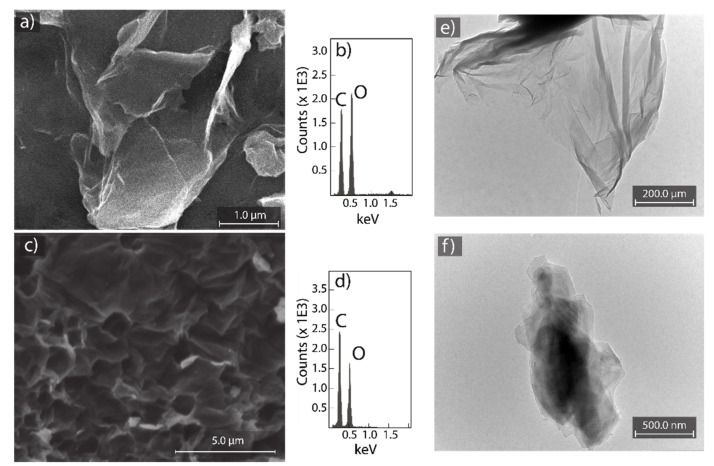
SEM morphology of (**a**) GO and (**c**) rGO. EDS measurements of (**b**) GO and (**d**) rGO. TEM images of (**e**) GO and (**f**) rGO.

**Figure 3 nanomaterials-12-00309-f003:**
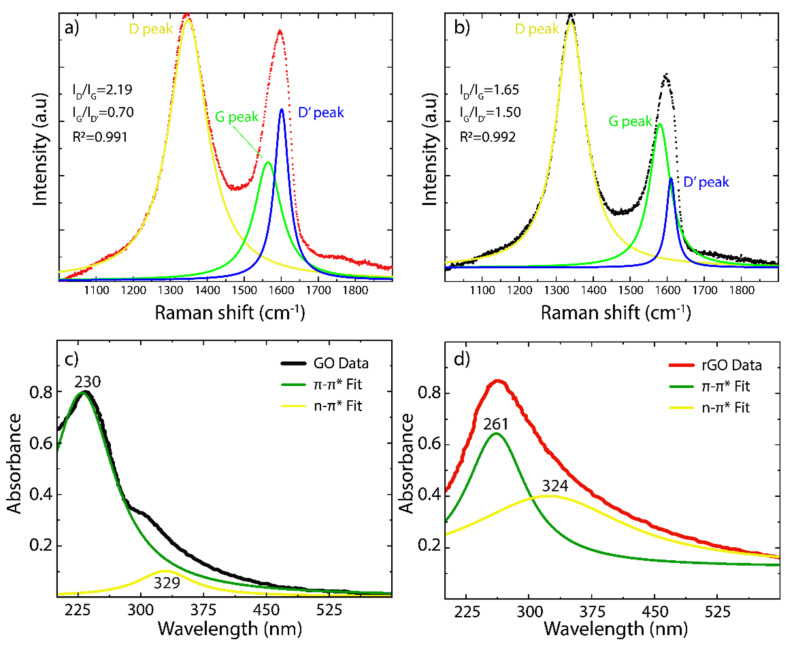
Raman spectra of (**a**) GO and (**b**) rGO from 1000 to 2000 cm^−1^. The intensity was normalized by the most intense peak. The Raman spectrum was fitted using Lorentzian functions. UV–Vis spectra of (**c**) GO and (**d**) rGO and the absorbance spectra were fitted by Lorentzian functions.

**Figure 4 nanomaterials-12-00309-f004:**
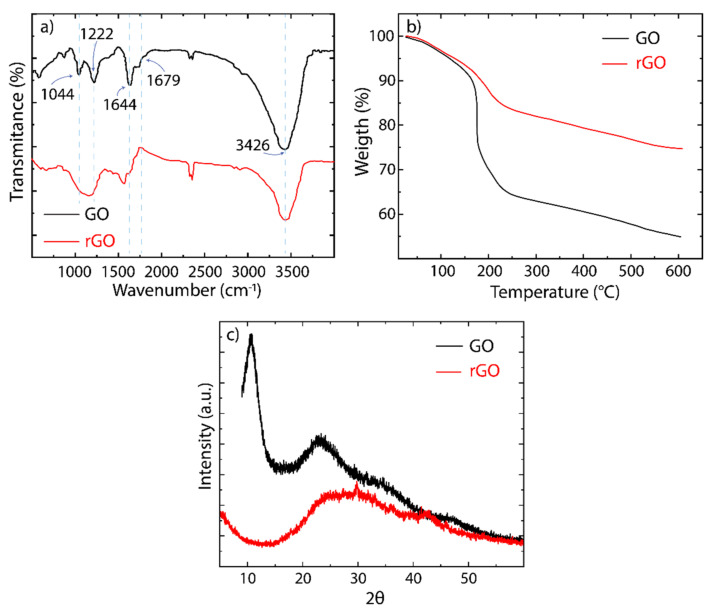
(**a**) Infrared spectra, (**b**) thermogravimetric study, and (**c**) XRD patterns of GO (black) and rGO (red), respectively.

**Figure 5 nanomaterials-12-00309-f005:**
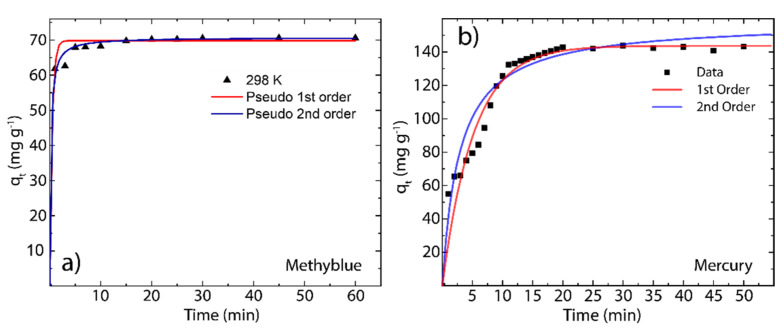
Adsorption kinetics of (**a**) MB on rGO and (**b**) Hg(II) on rGO as a function of contact time (60 min) at 298 K.

**Figure 6 nanomaterials-12-00309-f006:**
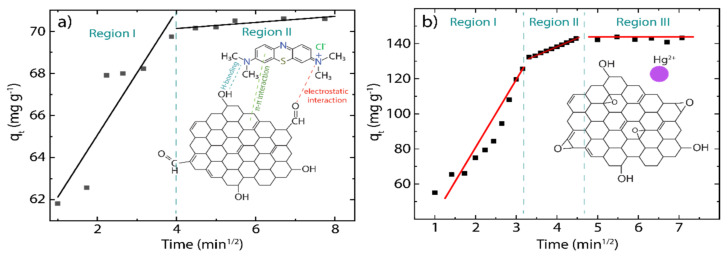
Intraparticle diffusion (IPD) study of (**a**) rGO+Mb and (**b**) rGO+Hg(II) at 298 K, showing different regions of linearity (MB concentration 100 mg L^−1^ and Hg(II) concentration 150 mg L^−1^).

**Figure 7 nanomaterials-12-00309-f007:**
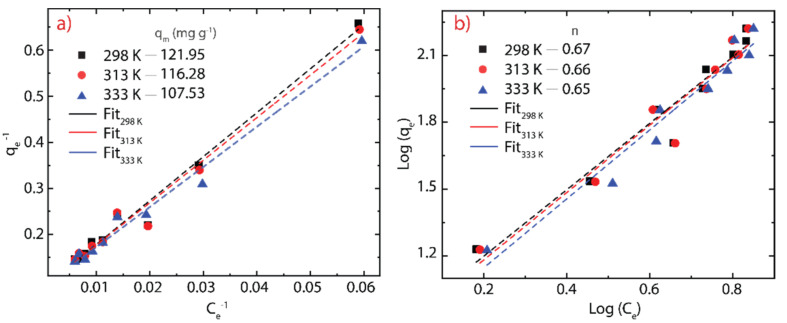
Adsorption isotherms of MB on rGO considering three different temperatures (289-333 K). (**a**) Langmuir model and (**b**) Freundlich model.

**Figure 8 nanomaterials-12-00309-f008:**
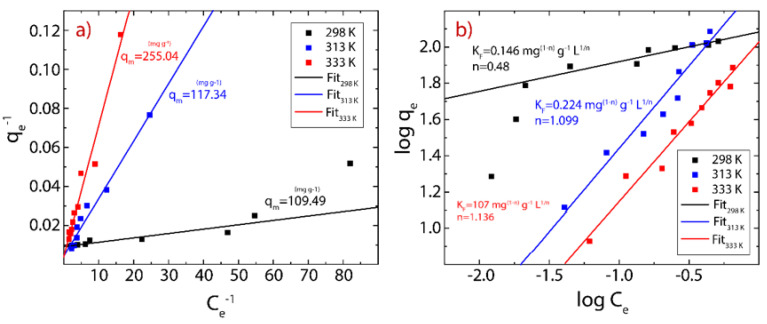
Adsorption isotherm of Hg(II) on rGO considering three different temperatures (289-333 K). (**a**) Langmuir model and (**b**) Freundlich model.

**Figure 9 nanomaterials-12-00309-f009:**
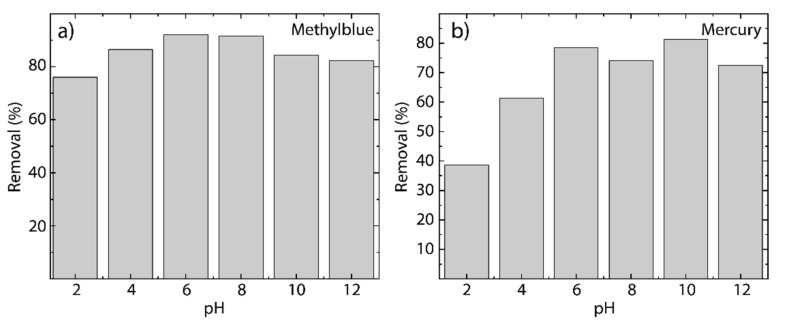
Removal percentage as a function of the pH (from 2 to 12) at 298 K of (**a**) MB on rGO and (**b**) Hg(II) on rGO (MB concentration 100 mg L^−1^ and Hg(II) concentration 100 mg L^−1^).

**Figure 10 nanomaterials-12-00309-f010:**
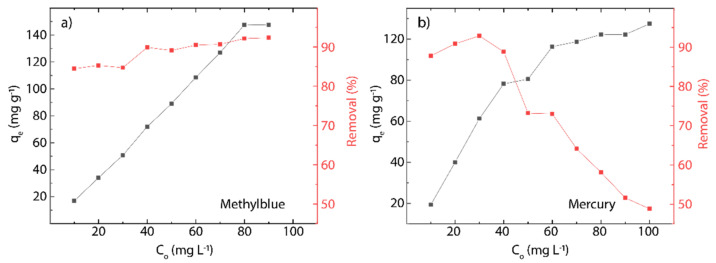
Effect of the initial concentration on the adsorption process at 298 K of (**a**) MB or rGO and (**b**) Hg(II) on rGO. Adsorption capacity (black markers) and removal percentage (red markers).

**Figure 11 nanomaterials-12-00309-f011:**
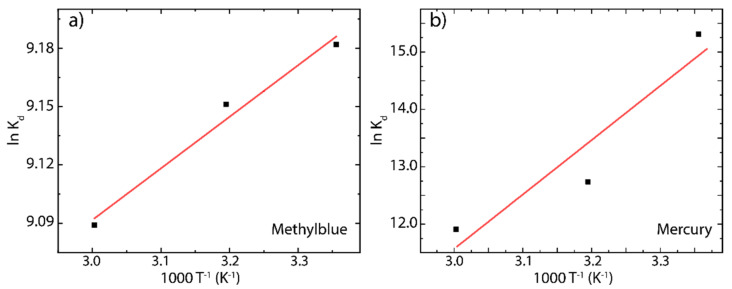
Van’t Hoff study for the adsorption of (**a**) MB on rGO and (**b**) Hg(II) on rGO.

**Table 1 nanomaterials-12-00309-t001:** Estimated parameters at 298 K of the pseudo-first-order model and the pseudo-second-order model.

Parameters	MB	Hg(II)
q_e_(exp) (mg g^−1^)	68.21	142.26
**Pseudo-first-order model**		
q_e_ (mg g^−1^)	69.82 ± 0.05	143.70 ± 5.70
k_1_ (min^−1^)	2.166 ± 0.03	0.19 ± 0.03
SSE	13.38	1826
R^2^	0.997	0.949
RMSE	1.157	8.546
**Pseudo-second-order model**		
q_e_ (mg g^−1^)	70.72 ± 0.08	158.30 ± 9.45
k_2_ (g mg^−1^ min^−1^)	0.075 ± 0.005	0.002 ± 0.001
SSE	5.239	2480
R^2^	0.999	0.931
RMSE	0.724	9.960

**Table 2 nanomaterials-12-00309-t002:** Estimated parameters of the intraparticle diffusion (IPD) model at 298 K.

	MB	Hg(II)
Parameters	Value	Value
Kp (mg g^−1^ min^−1/2^)	2.95 ± 0.67	7.82 ± 1.25
C (mg g^−1^)	59.17 ± 1.75	44.28 ± 7.75
Ri	0.132	0.69
R^2^	0.829	0.963

**Table 3 nanomaterials-12-00309-t003:** Parameters of the Langmuir and Freundlich models for the adsorption isotherms of MB onto rGO considering three different temperatures.

T (K)	Langmuir Model			Freundlich Model		
	K_L_ (L g^−1^)	q_m(cal)_ (mg g^−1^)	R^2^	K_F_ (mg^(1-n)^ g^−1^ L^1/n^)	n	R^2^
298	0.079	121.95	0.982	7.956	0.671	0.945
313	0.081	116.28	0.980	7.568	0.661	0.936
333	0.082	107.53	0.984	6.869	0.646	0.955

**Table 4 nanomaterials-12-00309-t004:** Parameters of the Langmuir and Freundlich models for the adsorption isotherms of Hg(II) onto rGO considering three different temperatures.

T (K)	Langmuir Model			Freundlich Model		
	K_L_ (L g^−1^)	q_m(cal)_ (mg g^−1^)	R^2^	K_F_ (mg^(1-n)^ g^−1^ L^1/n^)	n	R^2^
298	0.047	109.493	0.934	0.146	0.480	0.918
313	0.017	217.344	0.973	0.224	1.099	0.947
333	0.006	255.037	0.968	0.107	1.136	0.956

**Table 5 nanomaterials-12-00309-t005:** Thermodynamics parameters for Mb and Hg(II) adsorption on rGO at three different temperatures.

T (K)	$∆G^{0}(\mathrm{kJ}\;{\mathrm{mol}}^{-1})$	$∆H^{0}(\mathrm{kJ}\;{\mathrm{mol}}^{-1})$	$∆S^{0}(\mathrm{kJ}\;{\mathrm{mol}}^{-1}K^{-1})$
		MB	
298	−22.75	−2.20	0.069
313	−23.81		
333	−25.16		
		Hg(II)	
298	−39.84	−0.14	0.079
313	−31.55		
333	−32.97		

**Table 6 nanomaterials-12-00309-t006:** Comparative adsorption capacity of several adsorbents for the removal of MB and Hg(II).

MB		Hg(II)	
Adsorbents	Adsorption Capacity (mg g^−1^)	Adsorbents	Adsorption Capacity (mg g^−1^)
Graphene/SrAl2O_3_:Bi^3+^ [56]	42.93	GONR [35]	33.02
ß-cyclodextrin/MGO [57]	93.97	S-GO [36]	3490
g-C_3_N_4_ (Urea) [58]	2.51	GO-TSC [37]	231
Magnetic carboxyl functional nanoporous polymer [59]	57.74	S-doped g-C_3_N_4_/LGO [38]	46
Ag-Fe_3_O_4_-polydopamine [60]	45.00	GSH-NiFe_2_O_4_/GO [39]	272.94
Citrus hystrix-rGO [61]	276.06	HT-rGO-N [40]	75.8
This work	121.95	This work	109.49

## Data Availability

Not applicable.

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
