# Peer review of "Cationic Pollutant Removal from Aqueous Solution Using Reduced Graphene Oxide"

_nanomaterials, 2022, doi:10.3390/nano12030309_

Round 1
Reviewer 1 Report
The current manuscript entitled “Cationic Pollutant Removal from Aqueous Solution Using Reduced Graphene Oxide” by Tene et al. brought the materialistic aspects for wastewater treatment technologies. Presents findings for cationic pollutant adsorbent materials based on oxidized graphenes, providing a new perspective for removing methylene blue molecules and Hg(II) ions. The results and discussion section seem good. The manuscript was written well and can be accepted after addressing the following comments.
Minor Comments:
- It will be useful for the readers of “nanomaterials” if you provide more detailed information on lines 146-148.
- Improve the quality of Figures.
- “Discussions” should be “Discussion
- Notate the respective peaks in the “Infrared spectra” figure.
- Some potential literature is dedicated to the on-site detection of toxic pollutants and hazardous constituents that can provide a basic idea; those must be cited.
https://doi.org/10.1016/j.ccr.2021.214305
https://doi.org/10.1016/j.ccr.2021.214061
Author Response
Response to Reviewer 1 Comments
We thank the referee for his/her very valuable comments, which have indeed helped us improving the manuscript. We also thank the referee for considering that our work is interesting and well presented to be accepted for publication in Nanomaterials. We have carefully examined the referee’s comments and suggestions:
- It will be useful for the readers of “nanomaterials” if you provide more detailed information on lines 146-148.
We have provided a detailed description on preparing samples for characterization. Please see the revised version highlighted in yellow.
- Improve the quality of Figures.
Thank the Referee for this suggestion. As the figures lose resolution/size when compiling them directly in the Word template, we always provide Figures in *.pdf and *.eps format in the production phase as the article is accepted.
- “Discussions” should be “Discussion
The word has been corrected (highlighted in yellow).
- Notate the respective peaks in the “Infrared spectra” figure.
Figure 4a has been resized and labeled correctly to show the prominent peaks.
- Some potential literature is dedicated to the on-site detection of toxic pollutants and hazardous constituents that can provide a basic idea; those must be cited.
https://doi.org/10.1016/j.ccr.2021.214305
https://doi.org/10.1016/j.ccr.2021.214061
Thank the Referee for the suggested references, they have been briefly commented in the revised manuscript and appear as Refs 16 and 17 (highlighted in yellow).
Reviewer 2 Report
This paper can be accepted after moderate modifications:
(1) The introduction part is too broad. It is strongly suggested to combine the first three paragraphs into 1 shorter paragraph.
(2) It is strongly suggested to show the adsorption behaviour difference between GO and rGO. Better to add a comparison between your sample with others.
(3) The preparation of GO and rGO is the traditional way. The description should be deleted or modified "Most importantly, its synthesis follows an eco-friendly and cost-effective preparation method [32]. Until very recently, the synthesis of GO (oxidation, reduction, and purification) was considered a bottleneck when using rGO for large-scale water treatment technologies".
(4) Please cite the reference "Polymers, 2021, 13 (13), 2137".
Author Response
Response to Reviewer 2 Comments
We thank the referee for a careful reading of the manuscript and for his/her very valuable comments, which have indeed helped us improving the manuscript. We also thank the referee for considering that our work can be considered eligible to be published in Nanomaterials after moderate modifications:
- The introduction part is too broad. It is strongly suggested to combine the first three paragraphs into 1 shorter paragraph.
Thank you for this very important suggestion. We have modified and assembled paragraphs 1 and 2 into one as much as possible without losing the main idea and showing the non-proficient readers the problem we are trying to solve with our work and as-made rGO. Paragraph 3 was kept separate, but shortened a bit and focusing on conventionally used materials versus novel-synthesized carbon-based sorbents, say, GO and rGO. We hope this satisfies your suggestion as we think that this makes the problem and the proposed solution clear.
- It is strongly suggested to show the adsorption behaviour difference between GO and rGO. Better to add a comparison between your sample with others.
Thank you for this crucial suggestion in order to demonstrate the effectiveness of the as-made rGO. To light this requirement, we have prepared a comparative table between our results and some recently reported, considering GO and rGO. Please see Table 6 in the revised manuscript and the corresponding discussion.
- The preparation of GO and rGO is the traditional way. The description should be deleted or modified "Most importantly, its synthesis follows an eco-friendly and cost-effective preparation method [32]. Until very recently, the synthesis of GO (oxidation, reduction, and purification) was considered a bottleneck when using rGO for large-scale water treatment technologies".
Thank you for this interesting comment. The paragraph has been modified/deleted as suggested. Please see in the revised manuscript highlighted in yellow.
- Please cite the reference "Polymers, 2021, 13 (13), 2137".
This reference has been commented in the revised manuscript (Introduction section, highlighted in yellow) and appears as Ref. 30.
Reviewer 3 Report
Paper is a concise study, generally well referenced and the introduction is a sufficient review. The experimental section is clear. The structure of GO and rGO were well characterized by spectroscopical and morphological characterization. It is better to add other adsorption capacity of rGO, GO and their derivatives in literature to compare the performance of the current rGO. Moreover, the desorption and recycle use of adsorbent should be discussed.
Author Response
Response to Reviewer 3 Comments
We thank the Referee for considering that our work “is a concise study, generally well referenced and the introduction is a sufficient review”. As well, “The experimental section is clear…” We have carefully considered the suggestions, which have indeed helped us improving the manuscript:
- It is better to add other adsorption capacity of rGO, GO and their derivatives in literature to compare the performance of the current rGO.
Thank you for this crucial suggestion to demonstrate the effectiveness of the as-made rGO. To meet this requirement, we have prepared a comparative Table between our results and some recently reported. Please see Table 6 in the revised manuscript.
- Moreover, the desorption and recycle use of adsorbent should be discussed.
Thanks to the Reviewer for this important comment. At the end of Section 3, we offer a detailed discussion on the reuse and desorption of rGO. Please see the final paragraph of section 3 in the revised manuscript highlighted in yellow.
Reviewer 4 Report
The article entitled as Intercalated water and electron trapping sites for a hole mediated 2 photocatalysis in self-decorated WO3 core-shell nanoribbons has scientific value and relevance. It is a timely manuscript. However, there are some gap in this. The following point must be considered before final submission
- How the reusability adsorbent was assured?
- The FTIR needs more explanation regarding formation of adsorbent.
3 The highlights must be re-written in more concise way.
- The GA must be improved.
- Can authors provide XRD of used photocatalyst
6 The conclusion must be re-written in more conclusive way
Author Response
Response to Reviewer 4 Comments
Respected Reviewer, thank you for a careful reading of the manuscript. We regret that the current comments and suggestions refer to other work. We would love to answer your questions and accept your comments/suggestions. If there is an opportunity to receive ours, we are confident that they will help us to substantially improve our manuscript.
Till then
The corresponding author
Round 2
Reviewer 3 Report
accept in present form.
Reviewer 4 Report
the revisions are ok